# Efficacy of *Streptomyces murinus* JKTJ-3 in Suppression of *Pythium* Damping-Off of Watermelon

**DOI:** 10.3390/microorganisms11061360

**Published:** 2023-05-23

**Authors:** Mihong Ge, Xiang Cai, Dehuan Wang, Huan Liang, Juhong Zhu, Guoqing Li, Xianfeng Shi

**Affiliations:** 1State Key Laboratory of Agricultural Microbiology, Key Laboratory of Plant Pathology of Hubei Province, Huazhong Agricultural University, Wuhan 430070, China; gemihong@wuhanagri.com; 2Wuhan Academy of Agricultural Sciences, Wuhan 430070, China

**Keywords:** watermelon, damping-off, *Pythium aphanidermatum*, *Streptomyces murinus*, biocontrol

## Abstract

Damping-off caused by *Pythium aphanidermatum* (*Pa*) is one of the most destructive diseases for watermelon seedlings. Application of biological control agents against *Pa* has attracted the attention of many researchers for a long time. In this study, the actinomycetous isolate JKTJ-3 with strong and broad-spectrum antifungal activity was screened from 23 bacterial isolates. Based on the morphological, cultural, physiological, and biochemical characteristics as well as the feature of 16S rDNA sequence, isolate JKTJ-3 was identified as *Streptomyces murinus*. We investigated the biocontrol efficacy of isolate JKTJ-3 and its metabolites. The results revealed that seed and substrate treatments with JKTJ-3 cultures showed a significant inhibitory effect on watermelon damping-off disease. Seed treatment with the JKTJ-3 cultural filtrates (CF) displayed higher control efficacy compared to the fermentation cultures (FC). Treatment of the seeding substrate with the wheat grain cultures (WGC) of JKTJ-3 exhibited better control efficacy than that of the seeding substrate with the JKTJ-3 CF. Moreover, the JKTJ-3 WGC showed the preventive effect on suppression of the disease, and the efficacy increased with increase in the inoculation interval between the WGC and *Pa*. Production of the antifungal metabolite actinomycin D by isolate JKTJ-3 and cell-wall-degrading enzymes such as β-1,3-glucanase and chitosanase were probably the mechanisms for effective control of watermelon damping-off. It was shown for the first time that *S. murinus* can produce anti-oomycete substances including chitinase and actinomycin D. This is the first report about *S. murinus* used as biocontrol agent against watermelon damping-off caused by *Pa*.

## 1. Introduction

Damping-off of cucurbits is a global disease in nursery beds, and seedling losses have been reported ranging from 5% to 80% in infected areas [1,2]. Several pathogens in the genera *Pythium*, *Rhizoctonia*, and *Fusarium* [3,4,5,6] are responsible for the damping-off disease, of which *Pythium* is the most economically destructive pathogen with a broad host range worldwide [7,8]. Once *Pythium* pathogens are localized in a certain area, these pathogens will rapidly attack the seeds and seedlings, resulting in seed softening and rotting at the pre-emergence stage and/or seedling damping-off at the post-emergence stage [9].

Varieties resistant to the pathogens of damping-off disease in cucurbits have seldom been reported. Fungicides are generally employed for chemical control against damping-off disease [10,11,12]. However, the overuse of fungicides has led to resistance of pathogens and has failed to control *Pythium* damping-off disease [13,14,15]. Moreover, overuse of chemical agents threatens human health and causes ecological problems. Therefore, there is growing interest in developing alternative control methods, including biological control. 

Beneficial microorganisms and their metabolites are used as biocontrol agents in biological control. Many naturally antagonistic bacteria and fungi have been used to suppress cucurbit *Pythium* damping-off [16,17,18,19]. Seed or substrate treatment with antagonistic bacteria or fungi is a good option for disease control. Previous results demonstrated that four actinomycete isolates mixed with the soil could significantly decrease the occurrence of cucumber damping-off disease, and their control effects were close to that of metalaxyl under pot culture [20]. Roberts et al. [21] reported the control effect of cucumber, melon, and pumpkin seed treatment with the cells and cell-free extracts of *Serratia marcescens* N4-5 against cucurbit *Pythium* damping-off. Huang et al. [22] also reported that the seed-coating treatment or soil-mixing treatment with *Trichoderma* effectively controlled damping-off disease. These biocontrol agents play a role in biocontrol by competition, production of antibiotics, inducement of systemic resistance, and mycoparasitism [23,24,25,26].

China’s cucurbit planting area ranks the first in the world, of which the main planted cucurbit is watermelon. The intensive seedling cultivation has played an important role in ensuring high yield and quality production of watermelon. *Pythium* damping-off is a disease that frequently occurs during watermelon seedling raising, and *P. aphanidermatum* (*Pa*) is the dominant pathogen [27,28]. In our previous study from 2016 to 2018, we found that cucurbit damping-off occurred in Hubei, China, with an incidence of 10% to 30% or even over 30%, especially in scions cultivated by trays for cucurbit grafting seedling raising. A total of 221 strains from diseased plant samples of *Pythium* damping-off in cucurbit seedlings were isolated, which were identified as *Pa*, *P. ultimum*, *P. irregular*, *P. spinosum*, and *Pythium* sp. Among these strains, *Pa* accounted for 52% of the total strains and became the predominant species in Hubei [29]. 

The current study investigated the biocontrol of watermelon damping-off caused by *Pa*. The objectives of the study were as follows: (i) to screen a broad-spectrum antagonistic isolate with biocontrol potential against damping-off disease caused by *Pa*; (ii) to identify the antagonistic isolate based on its morphological, cultural, physiological, biochemical, and taxonomic characteristics; (iii) to evaluate the prevention and control effects of seed and substrate treatment with different forms and different concentrations of the antagonistic isolate; and (iv) to reveal the potential biocontrol mechanisms of the antagonistic isolate.

## 2. Materials and Methods

### 2.1. Experimental Materials

A total of 36 microbial isolates collected in our previous laboratory work were used in this study, including 10 actinomycete isolates (F46-1, F54, F77, H55-1a, JKTJ-3, V10a, V61a, V61b, W143, W143-1), 13 bacterial isolates (JKTJ-1, JKTJ-11, JKTJ-2, MKCC1, MKCC2, MKCC3, MKCC4, B.amy1, B.amy2, Pse2-22, Pse2-21, Lat6-4, S16-1), and 13 plant pathogens (*Pa*, *Rhizoctonia solani*, *Colletotrichum gloeosporioides*, *Botrytis cinerea*, *Stagonosporopsis cucurbitacearum*, *Verticillium dahlia*, *Phomopsis vexans*, *Fusarium oxysporum* f.sp. *hiveum*, *Leptosphaeria biglobosa*, *Phomopsis asparagi, Fusarium solani, Colletotrichum capsic*, and *Sclerotinia sclerotiorum*). These isolates were stored in the Plant Protection Laboratory of Wuhan Academy of Agricultural Sciences, Wuhan, China. Bacterial isolates were grown on nutrient agar (NA, beef extract 3 g, peptone 10 g, NaCl 5 g, glucose 10 g, agar powder 13 g, water 1000 mL, pH 7.0 to 7.2), and actinomycete isolates and plant pathogens were grown on potato dextrose agar (PDA, peeled potato 200 g, glucose 20 g, agar powder 13 g, water 1000 mL). All isolates were incubated at 28 °C in the dark. Isolate JKTJ-3 was deposited at the China Center for Type Culture Collection with registration number CCTCC M 20211271. Watermelon (*Citrullus lanatus* var. Zaojia 8424) seeds were purchased from Xinjiang Mingxin Kehong Agricultural Technology Co., Ltd. (Urumqi, China) were used in this experiment.

### 2.2. Screening of Biocontrol Agents

The dual culture method was used to screen the antagonists from 23 actinomycete and bacterial isolates against *Pa* following the previously described method [30]. Briefly, a full ring of the candidate isolate spores were streaked on one side of the PDA plate (a diameter of 9 cm) with the spore line 2.5 cm away from the plate center, and cultured at 28 °C. After a 2-day cultivation, a 6 mm diameter fungal disc taken from the edge of a 2-day-old colony of *Pa* was placed on the center of a PDA plate and cultured. PDA plates without actinomycetes or bacteria were used as controls. All the plates were incubated at 28 °C for 48 h. When the control of *Pa* covered the whole plate, the radius of the inhibited growth of the pathogen in each plate was measured. The inhibition rate (*IR*) was calculated using the following formula:*IR* = 100 × (45 − *A*)/45
where *A* and 45 indicate the radius of the inhibited growth of the pathogen and the radius of the petri dish, respectively.

Further broad-spectrum tests of the screened actinomycete and bacterial isolates were carried out. The above-mentioned dual culture method was adopted with the slight modification that the pathogens and the antagonists were simultaneously inoculated. PDA plates without the screened antagonists were used as the control. All the plates were cultured at 28 °C. When the controls of phytopathogenic fungi covered the whole plate, the width of the inhibition zone in each plate was measured. The *IR* was calculated, as described above.

Considering that the antagonists screened in vitro by our previous study tended to lose their activities in vivo, our preliminarily screened antagonistic actinomycetes were rescreened. The methods reported by Paul et al. [31] and Li et al. [32] were used with some modification in this experiment. Spores were scraped off the surface of the 5-day-old colonies cultured at 28 °C on PDA plates. The spore concentration was determined with a hemocytometer and then adjusted to 1 × 10^7^ spores/mL. One mL of spore suspension of the antagonistic actinomycetes was inoculated into 100 mL of PDB (the same components as PDA excluding agar powder) and cultured at 28 °C in a 220 r·min^−1^ rotary shaker for 4 days. The obtained fermentation culture (FC) was used as the antagonistic actinomycete inoculum. The substrate composed of peat (particle size, 0–7 mm; pH, 5.6; electrical conductivity (EC), 0.23 dS/m; Floragard, Germany) and perlite (particle size, 3–6 mm) at the ratio of 3:1 (*v/v*) was sterilized by autoclaving at 121 °C for 30 min. The mixture of the sterilized substrate and antagonistic actinomycete inoculum at the ratio of 5:1 (*v/v*) was used as the potting substrate, and put into pots (9 cm × 8 cm× 6 cm) at 200 mL per pot. Then a 5 mm diameter mycelium disc of *Pa* (2-day-old) was placed at the depth of 1.5 cm below the center of the potting substrate. The watermelon seeds were surface sterilized in 1% (*w/v*) formaldehyde for 1 h and rinsed with tap water, followed by 8 h soaking in sterile water. Then the seeds were wrapped with a towel and germinated at 28 °C for 36 h. Germinated seeds with about 1 cm of radicle length were selected, spread to each pot, and covered with the potting substrate (about 1.0 cm thick). The sterile substrate was mixed with the 300-fold diluted solution of 70% hymexazol wettable powder (WP) (Shanxi Biaozheng Crop Science Co., Ltd., Weinan, China) and sterile water, respectively, with the latter used as controls. Each treatment was carried out in triplicates with 5 pots per replicate and 20 seeds per pot. All the pots were covered with a layer of plastic film to retain the moisture, and the seeds were incubated at 30 °C during the pre-emergence stage and then cultivated during the post-emergence stage in a climate chamber (25 °C for 12 h light/18 °C for 12 h dark photoperiod, 90% humidity). The number of seed emergence was recorded at day 5 after sowing, and the number of the damping-off seedlings was counted at day 7 after sowing. The seedling damping-off incidence (*DI*) and seedling rate (*SR*) were calculated using the following formulas [18]:*DI* = 100 × *B*/20
*SR* = 100 × (*A* − *B*)/20
where *A* and *B* indicate the total number of emerged seeds and the number of seedlings with damping-off, respectively.

### 2.3. 16S rDNA Sequencing

One mL of isolate JKTJ-3 spore suspension was inoculated into 100 mL PDB and cultured in a 220 r·min^−1^ rotary shaker for 48 h. Subsequently, 2 mL of the cell suspension was centrifuged at 10,000 r·min^−1^ for 5 min, and the supernatant was discarded. The cells were resuspended in sterile water and washed twice. The genomic DNA of isolate JKTJ-3 was extracted with a bacterial genomic DNA extraction kit (Tiangen Biochemical Technology Co., Ltd., Beijing, China) according to the manufacturer’s instructions. The DNA extract was stored at −20 °C. Subsequently, 16S rDNA gene was amplified by PCR in 25 μL system containing 1 μL PCR Mix (Aidlab Biotechnologies Co., Ltd., Beijing, China), 1 μL of each primer (10 pmol/μL, Tsingke Biotechnology Co., Ltd., Beijing, China), 0.5 μL template DNA (75 μg/mL), and 21.5 μL deionized water. The universal primer pair 27f (5′-AGA GTT TGA TCC TGG CTC AG-3′) and 1495r (5′-CTA CGG CTA CCT TGT TAC GA-3′) was used. The PCR was conducted as follows: pre-denaturation at 94 °C for 5 min, 30 cycles of denaturation at 94 °C for 1 min, annealing at 58 °C for 1 min, and extension at 72 °C for 2 min, followed by a final extension at 72 °C for 10 min. The PCR product was detected by 1% (*w/v*) of agarose gel electrophoresis. Afterwards, the target DNA fragments were recovered with a DNA Recovery Kit (Axygen AP-GX-50, Axygen Biotechnology (Hangzhou) Co., Ltd., Hangzhou, China), and connected to the pMD^®^ 18-T vector (Takara Biotechnology (Dalian) Co., Ltd., Dalian, China). Finally, the recombinant vectors were transformed into the competent cells of *Escherichia coli* JM109 (Takara Biotechnology (Dalian) Co., Ltd., Dalian, China), and the positive clones containing the target DNA fragments were sequenced (Tsingke Biotechnology Co., Ltd., Beijing, China). The 16S rDNA sequence of isolate JKTJ-3 was aligned against the EzBioCloud server and the GenBank databases to obtain the homology sequences. The multiple sequence alignment was performed by the CLUSTAL W program, and a phylogenetic tree was constructed by a maximum-likelihood method using MEGA X software (Version 10.0.5). The support of each clade was determined by a bootstrap analysis with 1000 replications [33]. A matrix of pairwise distances was generated using the Tamura–Nei model [34].

### 2.4. Determination of Morphological, Cultural, and Physiological Features

The morphological characteristics of aerial hypha, spore hypha, and spore were observed by scanning electron microscope (SEM, Carl Zeiss Co., Ltd., Oberkochen, Germany). The cultural characteristics of isolate JKTJ-3 on the ISP medium were determined as previously reported [35,36]. Specifically, the aerial spore mass color, substrate mycelium pigmentation, and soluble pigment production of isolate JKTJ-3 were recorded on yeast extract–malt extract agar (ISP-2), oatmeal agar (ISP-3), inorganic salts–starch agar (ISP-4), glycerol–asparagine agar (ISP-5), and peptone–yeast extract–iron agar (ISP-6), respectively. The pigment presence on Bennett’s agar (BM) [37] and Gause’s No. 1 [38] was also investigated. The growth characteristics of isolate JKTJ-3 were observed after 7-day incubation at 28 °C under dark conditions.

Carbon source utilization, growth temperature, NaCl tolerance, and pH sensitivity of isolate JKTJ-3 were determined. A total of 8 carbon sources were used including D-glucose, sucrose, D-fructose, D-xylose, L-rhamnose, raffinose, L-arabinose, and D-mannitol. These carbon sources were prepared in 10% (*w/v*) solution and sterilized through a 0.22 μm syringe filter (Beijing Labgic Technology Co., Ltd., Beijing, China), and then mixed with ISP-9 medium to reach the final concentration of 1% (*m/v*) in the plate. The growth of isolate JKTJ-3 in ISP-2 medium was examined at the temperature of 10–45 °C and NaCl concentration of 0.5–8% (*w/v*). The sensitivity of isolate JKTJ-3 to pH (3.0–8.0) was investigated in ISP-2 medium adjusted by 0.2 mol/L K_2_HPO_4_-HCl and 0.2 mol/L KH_2_PO_4_-K_2_HPO_4_. The growth characteristics were also recorded at day 7 after incubation at 28 °C in dark.

### 2.5. Determination of Biocontrol Efficacy of Isolate JKTJ-3

The JKTJ-3 FC was prepared as described in Section 2.2. The FC was filtered with qualitative filter paper to obtain the crude filtrate. The crude filtrate was filtered with a 0.22 μm syringe filter to obtain the isolate JKTJ-3 cultural filtrate (CF). One mL of isolate JKTJ-3 spore suspension was inoculated into 100 g sterile wheat grain medium and incubated at 28 °C for 15 days to obtain the isolate JKTJ-3 wheat grain culture (WGC).

The control effects of different contents of JKTJ-3 CF or WGC on watermelon damping-off disease were examined. One liter of the sterile substrate (using the same in vivo screening preparation) was mixed with 10, 50, 100, 150, 200, 250, and 300 mL of the CF, respectively, and added with 290, 250, 200, 150, 100, 50, and 0 mL of sterile water, respectively, to ensure consistent water content in the substrate. Similarly, one liter of sterile substrate was mixed with 10, 20, 40, 80, and 160 g of JKTJ-3 WGC, respectively, to ensure 30% of the water content in the substrate. One liter of the sterile substrate was mixed with 300 mL of 300-fold diluted solution 70% hymexazol WP as chemical treatment group, while one liter of sterile substrate was mixed with 300 mL of sterile water as control group. Into each pot was added 200 mL substrate. Pathogen inoculation and seed potting were performed, as described in Section 2.2. The damping-off incidence (*DI*) and seedling rate (*SR*) were determined, as described in Section 2.2. The seed emergence rate (*ER*) and control efficacy (*CE*) were calculated according to the following formulas:*ER* = 100 × *A*/20
*CE* = 100 × (*DI_SW_* − *DI*)/*DI_SW_*
where *A* represents the total number of seed emergence and *DI_sw_* represents the damping-off incidence under the treatment of sterilized water (*SW*).

The preventive effect of isolate JKTJ-3 CF and WGC on watermelon damping-off by substrate treatment was investigated. The JKTJ-3 CF and WGC was mixed with the sterilized substrate at the ratio of 3:10 (*v/v*) and 7.5:100 (*w/v*), respectively. All the substrates for WGC treatment were added to 300 mL of sterile water to ensure the 30% water content in substrates. The 300-fold diluted solution 70% hymexazol WP and SW were, respectively, mixed with the sterile substrate at the ratio of 3:10 (*v/v*) and used as chemical treatment group and control group, respectively. The 200 mL sterile substrate was put into each pot. Each pot was sown with 20 seeds (disinfected in 1% formaldehyde for 1 h, soaked for 8 h), and then covered with a layer of about 1.0 cm thick substrate. A 5 mm diameter mycelium disc of *Pa* (48 h old) was inoculated to about 2.5 cm depth below the center of the substrate at day 0, 1, 2, and 3 after sowing, respectively. Each treatment was repeated 3 times with 5 pots in one replicate. Seedling indices (ER, DI, SR, and RCE) were determined, as described above.

The disinfected seeds were soaked in FC or in CF for 4 h, 8 h, and 12 h, respectively. Seeds treated with sterile water for 4 h, 8 h, and 12 h were used as controls. The 200 mL sterilized substrate was put into pots. A 5 mm diameter mycelium disc of *Pa* was inoculated to about 1.5 cm depth below the center of the substrate. Every pot was sown with 20 soaked seeds and then covered with a layer of 1.0 cm thick sterilized substrate. Seedling indices (ER, DI, SR, and RCE) were detected as described above.

### 2.6. Determination of β-1,3-Glucanase and Chitosanase Activities Produced by JKTJ-3

One mL spore suspension (1 × 10^7^ spores/mL) of isolate JKTJ-3 was inoculated into 100 mL PDB, chit-PDB (PDB added with mycelial homogenate of *Pa*, 25 g/L), SDM (soluble starch, 10 g/L; K_2_HPO_4_, 0.3 g/L; MgCO_3_, 1 g/L; KNO_3_, 1 g/L; and NaCl, 0.5 g/L), and chit-SDM (SDM added with mycelial homogenate of *Pa*, 25 g/L), respectively. PDB, chit-PDB, SDM, and chit-SDM added with no spore suspensions were used as controls. The spore suspension of isolate JKTJ-3 was incubated in the abovementioned 4 media in 220 r·min^−1^ shaker at 28 °C for 4 days to obtain 4 fermentation cultures (FCs), and these 4 FCs were filtered with 0.22 μm syringe filters. The activities of the β-1,3-glucanase and chitinase of the resultant filtrates were, respectively, determined with a glucanase activity assay kit (AKSU038C, Beijing Boxbio Sci. & Technol. Co., Ltd., Beijing, China) and chitinase activity assay kit (AKSU045C) according to the manufacturer’s instructions. The two enzyme activities were expressed as U/mL. Each treatment was performed in triplicates.

### 2.7. Determination of Antifungal Metabolites Produced by JKTJ-3

The CF was put through a 0.22 μm syringe filter, and the filtrate was extracted with equal-volume ethyl acetate and vacuum-dried at 40 °C to obtain colloidal crude extract [39]. As previously described [6,40], the crude extract was analyzed by Waters 2695 ultra performance liquid chromatography (UPLC) coupled with Waters Micromass Quattro Micro^®^ mass spectrometry in positive and negative ion mode with a heated electrospray ionization (ESI). Finally, the UV absorption spectrum and mass spectrum of the active substances in crude extract was analyzed by LC-MS software (Masslynx version 4.1). The obtained information was input into the Chapman compound database (version 2003) to obtain the chemical components and structures of active substances in the crude extract.

### 2.8. Statistical Analysis

All data were processed and analyzed with WPS Excel 2019, and then Duncan’s new complex difference method in SAS v.9.0 (SAS Institute Inc., Cary, NC, USA) was used to conduct the analysis of variance (ANOVA), and the significant differences between different treatments in each experiment were compared at the level of *p* < 0.05.

## 3. Results

### 3.1. In Vitro Screening of Antagonists

After 48 h culture, 13 candidate bacterial isolates did not form any inhibition zone, whereas 10 candidate actinomycete isolates exhibited an inhibitory effect on *Pa* (Figure 1). Among them, actinomycete isolate JKTJ-3 displayed the strongest antagonistic effect on *Pa*, with an average inhibition rate of 96.4% (Figure A1), followed by isolates V61b, V10a, and H55-1a. The inhibition rates of these four isolates were significantly higher than those of the other six isolates (*p* < 0.05). Therefore, the actinomycete isolates JKTJ-3, V61b, V10a, and H55-1a were selected for the subsequent determination of the antifungal spectrum. The isolate JKTJ-3 had the strongest inhibitory effect against 12 pathogenic fungi with inhibition rates ranging from 69.3% to 95.1% and an average inhibition rate of 80.6% (Table 1, Figure A2). The average inhibition rate of isolate V61b was 67.6%, which was inferior to that of isolate JKTJ-3. The antagonistic effects of actinomycete isolates V10a and H55-1a were relatively weak. Therefore, actinomycete isolates JKTJ-3 and V61b were selected for subsequent rescreening in vivo.

### 3.2. In Vivo Screening of Biocontrol Agents

The results showed an evident decrease in the damping-off incidence under isolate JKTJ-3 treatment, which was significantly lower than that under hymexazol and V61b treatments (*p* < 0.05) (Figure 2A and Figure A1). Moreover, the seedling rate (*SR*) of isolate JKTJ-3 was also significantly higher than those of hymexazol and V61b treatments (*p* < 0.05) (Figure 2B). These results suggested that isolate JKTJ-3 had the potential for the biocontrol of watermelon damping-off caused by *Pa*, and thus this isolate was chosen for subsequent experiments.

### 3.3. Taxonomic Identification of Isolate JKTJ-3

Isolate JKTJ-3 grew normally on media of ISP-1–6, GS-1, and BM (Table 2). The aerial spore mass colors of this isolate presented gray-white, smoky-pink (Figure 3A), and gray-brown on ISP-2–5 and BM media, but no aerial spore mass was formed on ISP-1, ISP-6, and GS-1 media. The substrate hyphae of isolate JKTJ-3 were relatively straight with few branches (Figure 3B), presented yellow and orange, and produced yellow and orange soluble pigment. The spore chains of this isolate were in dense spiral form (Figure 3C) and its spores were short rod-shaped or subglobose with a diameter of less than 1 µm and surface ridges (Figure 3D). The isolate JKTJ-3 made use of D-glucose, sucrose, D-fructose, D-xylose, L-rhamnose, raffinose, L-arabinose, and D-mannitol, and this isolate could grow at 8% NaCl (*w/v*) or less. It grew at the minimum pH of 3.5 with temperature range of 15–43 °C. Other cultural, physiological, and biochemical characteristics are shown in Table 2. The morphological and physiological features appeared similar to those of *S. murinus*.

To investigate the taxonomic characteristics of isolate JKTJ-3, the fragment of 16S rDNA was sequenced, and the obtained nucleotide sequence was submitted to the GenBank database (accession number: OK271440). The 26 type isolates with high homology and 1 outgroup isolate were downloaded from the NCBI database, and a phylogenetic tree was constructed based on their 16S rDNA gene alignments with MEGA X software (Figure 4). The phylogenetic analysis demonstrated that isolate JKTJ-3 has a close relationship with *S. murinus* as well as with *S. costaricanus*, *S. graminearus*, and *S. phaeogriseichromatogenes.* Taking the results of morphological/physiological features and molecular phylogeny together, isolate JKTJ-3 more likely belongs to *S. murinus.*

### 3.4. Biocontrol Efficacy of JKTJ-3 by Treatment of Seeding Substrate

To reveal whether there were differences in the biocontrol efficacy between different forms and different contents of JKTJ-3′s active ingredients, biocontrol efficacy of different concentrations of JKTJ-3 CF and WGC were determined by substrate treatment. Biocontrol efficacy of isolate JKTJ-3 CF is shown in Figure 5. The emergence rates of watermelon seedlings (Figure 5A) increased and then decreased with the increase in the dosages of the JKTJ-3 CF from 10 mL/L to 300 mL/L, of which two dosages (100 and 150 mL/L) exhibited a higher emergence rate than the remaining dosages (*p* < 0.05). Seedling rate (Figure 5B) and seed emergence rate displayed a similar trend, but 150 mL/L and 200 mL/L CF treatments showed significantly higher seedling rate than other treatments (*p* < 0.05). The damping-off incidence of watermelon seedlings (Figure 5C) decreased and then increased with the increase in JKTJ-3 CF, of which 150 mL/L and 200 mL/L CF treatments exhibited a lower damping-off incidence than other treatments (*p* < 0.05). The biocontrol efficacy (Figure 5D) under 150 mL/L CF treatment was 51.4%, which was significantly higher than 45.1% under 200 mL/L CF treatment, and biocontrol efficacy of these two dosages (150 and 200 mL/L) was significantly better than other treatments (*p* < 0.05). In addition, the biocontrol efficacies of JKTJ-3 CF within the 50–200 mL/L concentration range were much better than that of hymexazol treatment (*p* < 0.05).

Differences in seed emergence rate, seedling rate, control efficacy, and damping-off incidence were observed between JKTJ-3 CF and WGC. The seed emergence rates (Figure 6A) under the WGC treatments from 10 g/L to 160 g/L were higher than 91%, and the 160 g/L WGC treatment exhibited the minimum seed emergence rate, but 10–160 g/L WGC treatments exhibited no significant difference from the control in seed emergence rate. Seedling rates (Figure 6B) increased as the concentrations of WGC increased. The seedling damping-off incidences (Figure 6C) decreased with the increasing concentrations of WGC. Under 160 g/L WGC treatment, the damping-off incidence and seedling rate were 6.7% and 84.9%, respectively, both of which were significantly different from those under other treatments (*p* < 0.05). The control efficacy of WGC treatment (Figure 6D and Figure A3) was within the range of 53.2% to 92.4%, which was significantly higher than that of hymexazol treatment (*p* < 0.05). The control efficacy of 160 g/L WGC treatment showed the best control efficacy against watermelon damping-off.

### 3.5. Protective Efficacy of JKTJ-3 by Treatment of Seeding Substrate

We further examined the protective effects of JKTJ-3 CF and WGC against *Pa* damping-off disease by substrate treatment. It can be seen from Table 3 that the emergence rate, seedling rate, and control efficacy of watermelon seedlings with JKTJ-3 CF by substrate treatment increased when the *Pa* inoculation interval time became longer, while damping-off incidence decreased. The lowest damping-off incidence, highest emergence rate, and highest seedling rate were 34.7%, 95%, and 62.0%, respectively, which was 40.7% lower, 13.7% higher, and 82.4% higher than that of simultaneous inoculation of JKTJ-3 CF and *Pa* (0 day interval), respectively. At the interval of 1–3 days, JKTJ-3 CF treatment exhibited the optimal damping-off incidence, seedling rate, and protective efficacy. Compared with that of fungicide hymexazol WP, the protective efficacy of JKTJ-3 CF at the interval of 1 d, 2 d, and 3 d increased by 25.8%, 122.3%, and 614.6%, respectively.

A similar change trend was observed in WGC substrate treatment (Table 4, Figure A4). The highest seed emergence rate and highest seedling rate were 96.7% and 94.3%, respectively, which was 9.5% higher and 36.3% higher than that of simultaneous inoculation of JKTJ-3 WGC and *Pa* (0 day interval), respectively. Moreover, no seedling damping-off was observed under WGC treatment at the interval of 3 days. The damping-off incidence under WGC treatment at different inoculation intervals was significantly lower than that under control treatment and hymexazol WP treatment, while the seedling rate was significantly higher (*p* < 0.05). In the interval of 0–3 days, the protective effects under WGC treatment at different intervals was 36.7–227.9%, significantly higher than that under hymexazol WP treatment (*p* < 0.05). These findings suggested that *Streptomyces* JKTJ-3 CF and WGC treatments had a good protective effect on watermelon damping-off, and WGC exhibited better effect than CF.

### 3.6. Biocontrol Efficacy of Isolate JKTJ-3 by Seed Treatment

Further, we investigated the potential biocontrol efficacy of *Streptomyces* JKTJ-3 FC (fermentation culture) and CF (cultural filtrate) by seed treatment (Table 5). Compared with the control, seed treatment with JKTJ-3 FC and CF significantly (*p* < 0.05) reduced the damping-off incidence, but increased the seedling rate. The biocontrol efficacy of CF seed treatment ranged from 32.1% to 50.7%, while that of FC treatment was 14.7–39.2%. In general, watermelon seed-soaking treatment with both JKTJ-3 CF and FC could protect watermelon seedlings against the pathogen *Pa*, but the control efficacy of JKTJ-3 CF was superior to that of JKTJ-3 FC.

### 3.7. Biocontrol Mechanisms of Isolate JKTJ-3

The activities of β-1,3-glucanase (Figure 7A) and chitinase (Figure 7B) were detected, and the results showed that after isolate JKTJ-3 was cultured in PDB medium and chit-PDB medium, respectively, the activity of β-1,3-glucanase in PDB medium was 6.39 U/mL, and that of chitinase in chit-PDB medium was 4.07 U/mL, which were higher than those in SDM medium and chit-SDM medium, respectively.

The active compounds of JKTJ-3 ethyl acetate crude extract were detected by LC-MS. The results showed that a compound at 6.62 min showed a strong ultra violet (UV) absorption peak (Figure 8A) corresponding to a characteristic absorption peak at 442 nm (Figure 8B). In the mode diagram of positive and negative ion flow, the two corresponding mass spectra peaks at 1255.64 *m/z* (Figure 8C) and 1256.29 *m/z* (Figure 8D) exhibited high abundances (M − 1 = 1255.64, M + Na (23) = 1256.29). According to the mass spectrum rules, the molecular weight of the compound was about 1256.0. The molecular weight and UV absorption peak wavelength of the active compounds were input into the Chapman Database, and a polypeptide antibiotic with similar properties to actinomycin D was obtained. The molecular formula and molecular weight of this compound were C_62_H_86_N_12_O_16_ and 1255.42, respectively, and a characteristic absorption peak was observed at the wavelength of 442 nm, which was consistent with the mass spectrometry analysis data. The above results indicated that actinomycin D was one of the active substances of JKTJ-3 ethyl acetate crude extract.

## 4. Discussion

Some studies have reported the application of microorganisms in the biological control of *Pythium* damping-off [20,41,42]. In this study, we obtained actinomycete isolate JKTJ-3 with the strongest antagonistic activity against important pathogenic groups causing watermelon damping-off such as *Pa*, *R. solani*, *F. oxysporum*, and *F. solani*. Our in vivo experiment data showed that the actinomycete isolate JKTJ-3 could prevent the watermelon seedlings from *Pa*-induced damping-off disease. According to Bac*Dive* (https://bacdive.dsmz.de), *S. murinus* was assigned as risk level 1 regarding biosafety, and it was strongly recommended for biological control agents. Thus, actinomycete isolate JKTJ-3 can be a potential biocontrol agent and deserves further study regarding its accurate identification.

It is difficult to accurately clarify isolate JKTJ-3 based on 16S rDNA alone, as one of the four reported species, including *S. costaricanus*, *S. murinus*, *S. graminearus*, and *S. phaeogriseichromatoge*. This result was in line with those of previous reports [43,44]. Furthermore, we separately discussed the morphological/physiological characteristics among four species. According to taxonomic characteristics described in *Manual of Streptomyces Identification* [38], the isolate *S. graminearus* does not produce soluble pigments in GS-1 medium, which was inconsistent with our findings of the isolate JKTJ-3, indicating that JKTJ-3 does not belong to *S. graminearus* species. Goodfellow et al. [45] have reported that the type isolate *S. phaeogriseichromatogenes* NRRL 2834 cannot grow at 40 °C and above, and it can use neither D-raffinose nor L-rhamnose as sole carbon sources, but our data showed that isolate JKTJ-3 could grow at 43 °C, and it did utilize D-raffinose and L-rhamnose, suggesting that JKTJ-3 does not belong to *S. phaeogriseichromatogenes* species. Esnard et al. [35] revealed that the type isolate *S. costaricanus* CR-43 (ATCC 55274) did not produce soluble pigments in ISP-3 medium or ISP-4 medium; it cannot use sucrose, L-arabinose, D-raffinose, or L-rhamnose, and it cannot grow in the presence of over 5% NaCl (*w/v*), which was quite different from our observation of isolate JKTJ-3, implying that JKTJ-3 does not belong to *S. costaricanus* species. As Esnard et al. [35] described, the type isolate *S. murinus* NRRL B-2286 can produce soluble pigments in ISP-2, ISP-3, ISP-4, and ISP-5 media, but not in ISP-6 medium; it can utilize sucrose, L-arabinose, cottonseed sugar, and D-rhamnose as its only carbon source; and it can grow at 7% NaCl (*w/v*), which was highly consistent with our findings of the isolate JKTJ-3. Based on the above comparison of the four species, isolate JKTJ-3 was identified as *S. murinus.*

As previously reported, it was confirmed that *S. murinus* could efficiently produce a new immobilized glucose isomerase, which could produce more than 10,000 kg syrup dry substance per kg enzyme under optimal industrial conditions [46]. Bandlish et al. found that *S. murinus* produce soluble and immobilized xylose (glucose) isomerases which could catalyze glucose to fructose [47]. Therefore, *S. murinus* contributes to the production of immobilized glucose isomerase, xylose isomerase, and others in food engineering. In addition, Fang et al. [48] reported that *S. murinus* has adenylate deaminase genes, which could become the gene source for constructing the constitutive expression of AMP deaminase. To the best of our knowledge, this is the first report of *S. murinus* in the biocontrol of watermelon *Pa* damping-off. Moreover, there are few studies on biocontrol of watermelon damping-off disease. *S. murinus* JKTJ-3 may become a new biocontrol agent against damping-off.

Many researchers refer to damping-off as a nursery disease which is usually associated with soil- or seed-borne pathogens [9]. Biological seed treatment and substrate treatment are two frequently used methods for controlling *Pythium* damping-off [49,50,51]. In this study, a comprehensive biocontrol strategy was adopted using *S. murinus* JKTJ-3 by seed treatment and substrate treatment with the FC, CF, and WGC. Watermelon seed-soaking treatment with *S. murinus* JKTJ-3 inhibited watermelon damping-off. The biocontrol efficacy was affected by the active compounds of JKTJ-3 and the soaking time. The biocontrol efficacy of the substrate treatment was better than that of the seed-soaking treatment. We also found that the biocontrol efficacy of *S. murinus* JKTJ-3 WGC was better than that of JKTJ-3 CF, which may be related to the continuous production of active compounds by the JKTJ-3 WGC.

In addition, the protective efficacy of the JKTJ-3 CF and WGC was improved as the *Pa* inoculation interval increased. As previously reported, the propagules of *Pythium* can propagate and colonize very rapidly under suitable temperature and humidity conditions, thus escaping natural antagonism [52]. Our data showed that the control efficacy of substrate treatments with *Pa* at 1–3 d inoculation interval was better than that at simultaneous inoculation (namely, 0 d inoculation interval). In other words, early inoculation of *S. murinus* JKTJ-3 active ingredient may exhibit an advantage in its colonization in the substrate, seeds, and seedling, or in inhibiting or killing pathogens before pathogens propagate massively, which eventually reduces the occurrence of watermelon damping-off disease. Thus, early inoculation of the biocontrol agent could increase the control efficacy of *S. murinus* JKTJ-3, which is consistent with findings reported by Becker and Schwinn [53].

Currently, there have been few reports on the biocontrol mechanism of *S. murinus*. Antagonism, as an action mechanism of controlling plant disease, mainly depends on antifungals to inhibit the growth and metabolism of pathogens [54,55]. Another molecular mechanism of antagonism is that a variety of enzymes inhibit, degrade, and hydrolyze other pathogens [56]. In this study, a large inhibition zone was found to be formed between the colonies of phytopathogenic fungi and isolate JKTJ-3, indicating that isolate JKTJ-3 could produce antibiotics. Our enzyme activity determination indicated that *S. murinus* JKTJ-3 did secrete β-1,3-glucanase and chitinase. β-1, 3-glucanase and chitinase activities can degrade the cell wall of pathogenic fungi, which has been identified as a main biocontrol mechanism in many studies [57]. Our LC-MC analysis confirmed that *S. murinus* JKTJ-3 produced actinomycin. D. Zajkowicz et al. [58] have reported that actinomycin D has antimicrobial and antiviral effects as well as anticancer and antitumor effects. Lei et al. [59] have revealed that *S. antibioticus* can strongly inhibit several pathogenic bacteria and fungi by producing actinomycin D. The results indicated that *S. murinus* JKTJ-3 produced multiple metabolites including chitinase, and actinomycin D, which might be part of the potential mechanism of isolate JKTJ-3′s biocontrol against watermelon damping-off disease. The specific functions of these metabolites in biocontrol remain to be further investigated.

Overall, *S. murinus* JKTJ-3 efficiently inhibited several damping-off pathogens of cucurbit crops in vitro, further confirming its ability to control *Pa* damping-off of watermelon. The isolate JKTJ-3 produced β-1, 3-glucanase, chitinase, and actinomycin D, which are probably the mechanisms for biocontrol of watermelon damping-off. *S. murinus* JKTJ-3 and its active ingredients as biocontrol factors are newly discovered, and the biocontrol mechanisms need to be further clarified.

## 5. Conclusions

This study revealed that the antagonistic *S. murinus* JKTJ-3 had a promising biocontrol potential in preventing and controlling cucurbit damping-off disease. We comprehensively evaluated the biocontrol effects of *S. murinus* JKTJ-3 by treating seeds of watermelon or seeding substrates for watermelon with *S. murinus* JKTJ-3 FC, CF, and WGC. Among three biocontrol treatments, *S. murinus* JKTJ-3 WGC treatment with substrates exhibited the optimal control efficacy. Our results indicated that *S. murinus* JKTJ-3 produced actinomycin D, β-1,3-glucanase, and chitinase, which might be part of the potential mechanisms for the JKTJ-3 biocontrol.

## Figures and Tables

**Figure 1 microorganisms-11-01360-f001:**
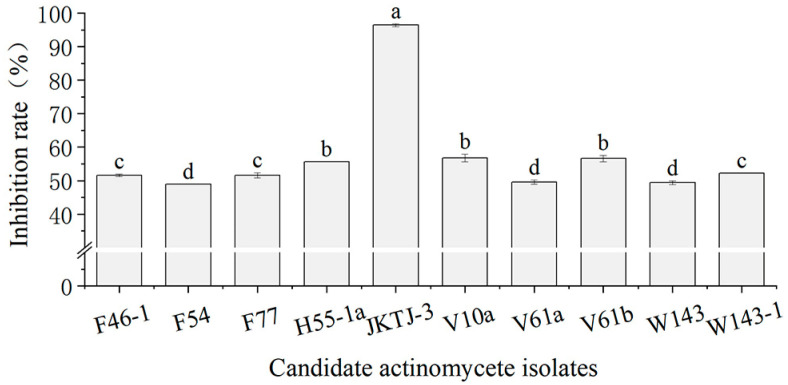
In vitro screening of 10 candidate actinomycete isolates against *Pa.* The different letters a–d were significantly different (*p* < 0.05).

**Figure 2 microorganisms-11-01360-f002:**
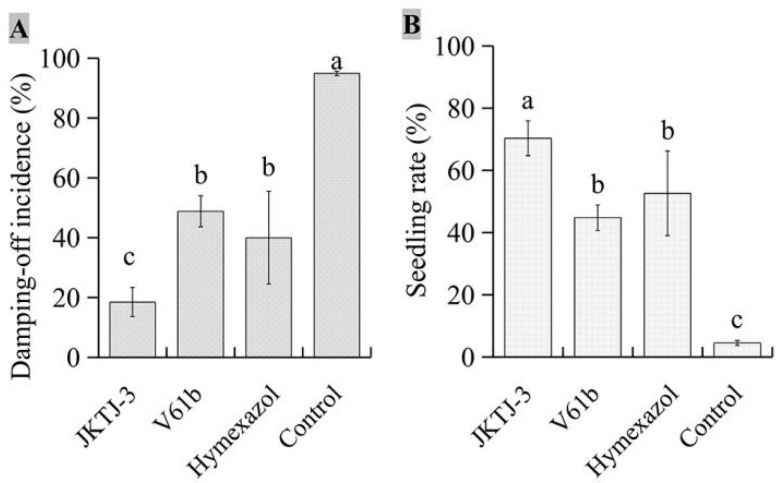
In vivo screening of two antagonistic actinomycete isolates against watermelon *Pa* damping-off: (**A**) damping-off incidence; (**B**) seedling rate. The different letters a–c were significantly different (*p* < 0.05).

**Figure 3 microorganisms-11-01360-f003:**
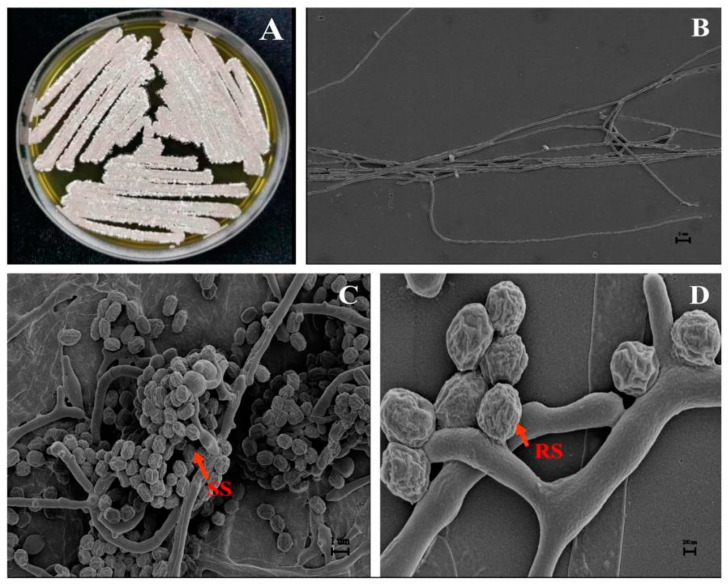
The morphological characteristics of aerial hypha, spore hypha, and spore of isolate JKTJ-3: (**A**) colony morphology (ISP-2, 28 °C, 7 d); (**B**) substrate mycelium (2 µm, SEM), few branches on the substrate mycelium; (**C**) spiral spore (SS) chains (1 µm, SEM); and (**D**) ridged spore (RS) (200 nm, SEM).

**Figure 4 microorganisms-11-01360-f004:**
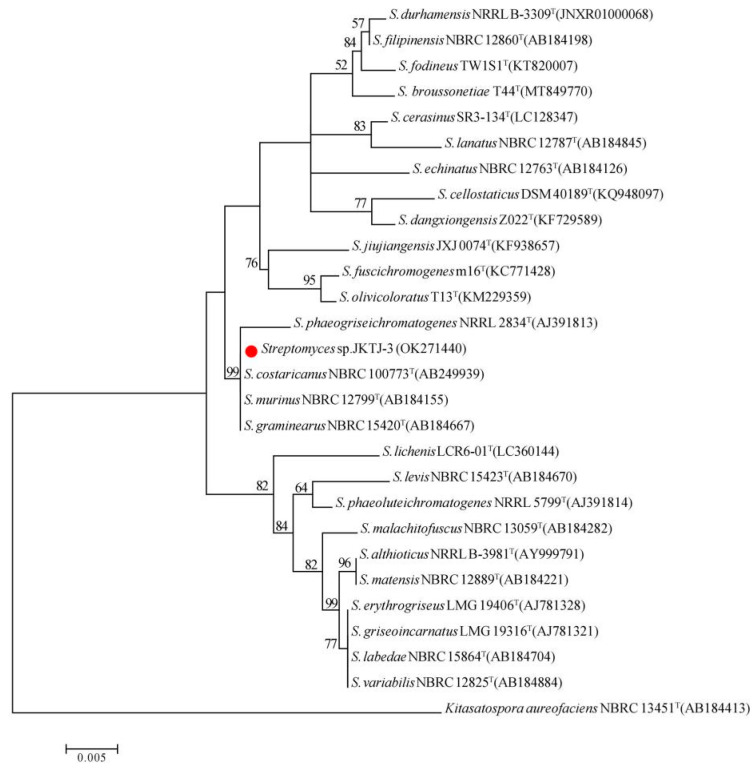
Phylogenetic tree inferred by the maximum-likelihood method based on 16S rDNA sequences. The bootstrap values of the maximum-likelihood analysis (*n* = 1000) over 50% are shown in the tree. The scale bar indicates 0.5% nucleotides variation per site. The red dot refers to the position of the isolate JKTJ-3 in the phylogenetic tree.

**Figure 5 microorganisms-11-01360-f005:**
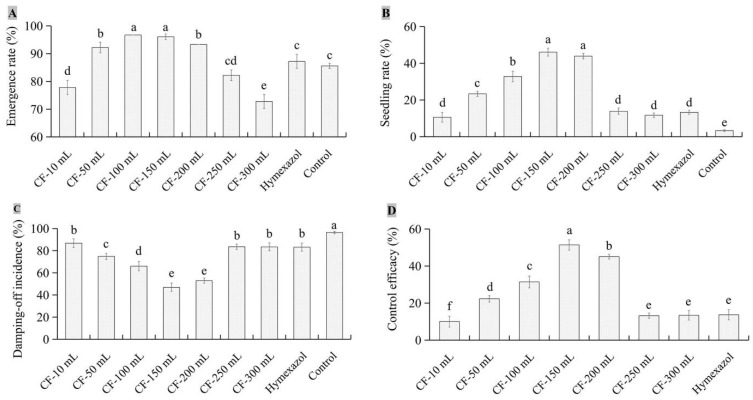
The biocontrol effects of different concentrations of isolate JKTJ-3 CF by treatment of seeding substrate: (**A**) emergence rate; (**B**) seedling rate; (**C**) damping-off incidence; and (**D**) control efficacy. The different letters a–f were significantly different (*p* < 0.05).

**Figure 6 microorganisms-11-01360-f006:**
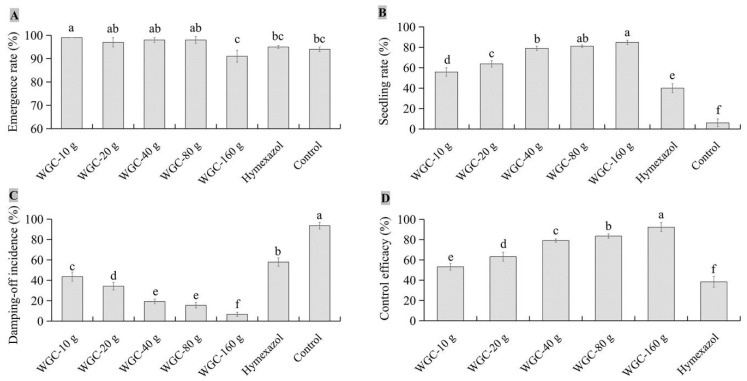
The biocontrol effects of different contents of isolate JKTJ-3 WGC by treatment of seeding substrate: (**A**) emergence rate; (**B**) seedling rate; (**C**) damping-off incidence; and (**D**) control efficacy. The different letters a–f were significantly different (*p* < 0.05).

**Figure 7 microorganisms-11-01360-f007:**
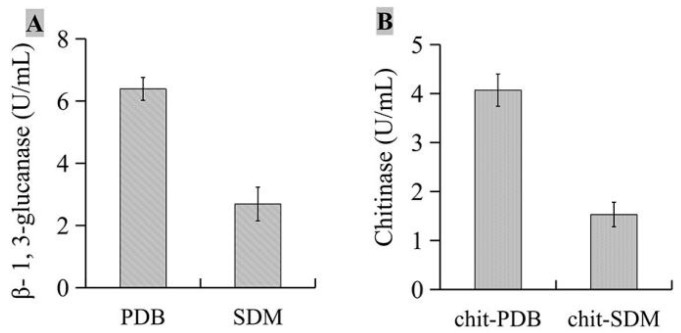
Analysis of β-1,3-glucanase (**A**) and chitinase (**B**) activities.

**Figure 8 microorganisms-11-01360-f008:**
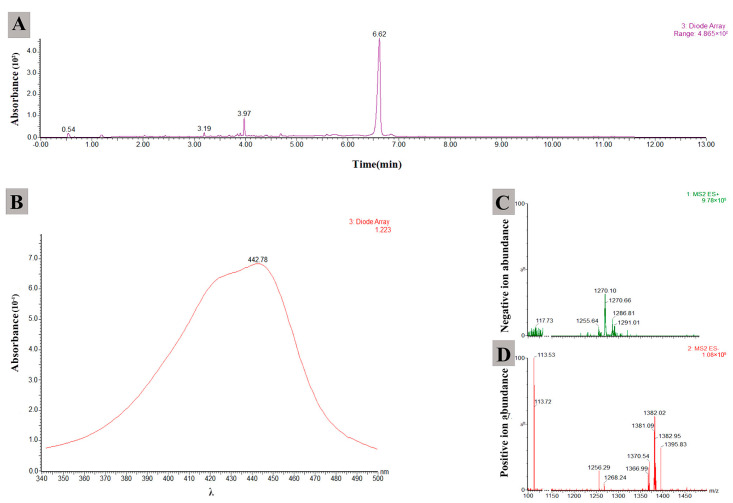
Analysis of active substances of isolate JKTJ-3 ethyl acetate extract by HPLC−MS: (**A**) HPLC−MS spectrum of ultraviolet absorption of JKTJ-3 ethyl acetate extract; (**B**) ultraviolet absorption spectrum of an active substance; (**C**) negative ion flow; and (**D**) positive ion flow.

**Table 1 microorganisms-11-01360-t001:** Comparison of four actinomycete isolates against 12 phytopathogenic fungi.

Pathogen	Inhibition Rate (%)			
	JKTJ-3	H55-1a	V10a	V61b
*Rhizoctonia solani*	91.1 ± 0.6 a ^1^	0.0 ± 0 c ^1^	0.0 ± 0 c ^1^	80.0 ± 0.4 b ^1^
*Stagonosporopsis cucurbitacearum*	69.6 ± 0.5 a	54.9 ±1.0 c	57.1 ± 1.1 c	65.8 ± 0.9 b
*Fusarium oxysporum* f. sp. *hiveum*	71.3 ± 0.4 a	0.0 ± 0 c	54.0 ± 0.8 b	54.7 ± 0.3 b
*F. solani*	76.7 ± 0.6 a	51.6 ± 1.2 c	53.8 ± 0.7 c	61.8 ± 1.5 b
*Botrytis cinerea*	69.3 ± 0.6 a	47.3 ± 0.8 c	59.3 ± 0.6 b	63.8 ± 1.1 b
*Colletotrichum gloeosporioides*	85.1 ± 0.9 a	56.7 ± 0.5 c	61.8 ± 0.9 b	63.3 ± 0.4 b
*C. capsic*	72.7 ± 0.3 a	55.1 ± 0.5 c	62.9 ± 0.6 b	64.0± 0.7 b
*Sclerotinia sclerotiorum*	84.0 ± 0.6 a	59.3 ± 0.6 c	0.0 ± 0 d	70.7 ± 0.8 b
*Verticillium dahliae*	95.1 ± 0.4 a	79.3 ± 0.7 b	64.0 ± 0.5 c	79.6 ± 1.2 b
*Leptosphaeria biglobosa*	91.8 ± 0.8 a	81.1 ± 0.4 b	66.2 ± 0.2 c	84.0 ± 0.6 b
*Phomopsis vexans*	78.2 ± 0.5 a	58.2 ± 1.1 c	50.7 ± 0.4 d	66.0 ± 0.8 b
*Phomopsis asparagi*	81.8 ± 1.1 a	0.0 ± 0 d	62.7 ± 0.6 b	57.8 ± 1.0 c

^1^ Means ± SD followed with the same letters within each column were not significantly different (*p* > 0.05).

**Table 2 microorganisms-11-01360-t002:** Cultural, physiological, and biochemical characteristics of isolate JKTJ-3.

Characteristic	Growth ^1^	Substrate Mycelia Color	Aerial Mycelia Color ^2^	Sporulation ^3^	Pigment
Growth characteristics					
ISP-1	++	Orange	−	NS	Orange
ISP-2	+++	Orange	GP/SP/DB	AS	Orange
ISP-3	+	Orange	W	SS	Yellow
ISP-4	++	Orange	W	SS	Yellow
ISP-5	+++	Orange	W/GP	AS	Orange
ISP-6	++	Orange	−	NS	Orange
GS-1	+	Yellow	−	NS	Yellow
BM	+++	Orange	W/GP/DB	AS	Orange
Utilization of carbon sources					
D-glucose	+	Not determined	Not determined	SS	Pale
Sucrose	++	Not determined	Not determined	SS	Pale
L-arabinose	+	Not determined	Not determined	SS	Pale
D-fructose	++	Not determined	Not determined	DS	Yellow
D-xylose	+++	Not determined	Not determined	SS	Yellow
D-mannitol	+++	Not determined	Not determined	SS	Yellow
Raffinose	+	Not determined	Not determined	SS	Pale
Rhamnose	++	Not determined	Not determined	SS	Pale
Growth response to NaCl					
NaCl (≤2%)	+++	Not determined	Not determined	AS	Orange to yellow
NaCl (5–8%)	+	Not determined	Not determined	NS	Yellow
NaCl (>8%)	−	Not determined	Not determined	NS	Pale
Growth response to pH					
pH 3.5	+	Not determined	Not determined	NS	Pale
pH 4–7	++/+++	Not determined	Not determined	SS	Yellow
pH 7.5–8.0	+	Not determined	Not determined	NS	Yellow

^1^ Growth: +, sparse mycelia; ++, dense mycelia; +++, highly dense mycelia; −, no growth. ^2^ Aerial mycelia color: −, no aerial hyphae; W, whitish; GP, grayish-pink; DB, dark-brown; SP, smoky-purple. ^3^ Sporulation: SS, sparse sporulation; DS, dense sporulation; AS, abundant sporulation; NS, no sporulation.

**Table 3 microorganisms-11-01360-t003:** Protective efficacy of isolate JKTJ-3 CF by treatment of seeding substrate.

Inoculation Interval	Treatment	Seed Emergence Rate (%)	Damping-Off Incidence (%)	Seedling Rate (%)	Control Efficacy (%)
0 d	JKTJ-3 CF	82.0 b ^1^	58.5 b ^1^	34.0 b ^1^	39.0 b ^1^
	Hymexazol WP	90.0 a	47.8 c	47.0 a	50.2 a
	Control	75.0 c	96.0 a	3.0 c	-
1 d	JKTJ-3 CF	90.0 a	40.0 c	54.0 a	57.5 a
	Hymexazol WP	90.0 a	51.1 b	44.0 b	45.7 b
	Control	86.0 b	94.2 a	5.0 c	-
2 d	JKTJ-3 CF	95.0 a	35.8 c	61.0 a	63.8 a
	Hymexazol WP	95.0 a	70.5 b	28.0 b	28.7 b
	Control	90.0 b	98.9 a	1.0 c	-
3 d	JKTJ-3 CF	95.0 a	34.7 c	62.0 a	63.3 a
	Hymexazol WP	95.0 a	86.3 b	13.0 b	8.9 b
	Control	95.0 a	94.7 a	5.0 c	-

^1^ The values followed with the same letters within each column for each assay were not significantly different (*p* > 0.05).

**Table 4 microorganisms-11-01360-t004:** Protective efficacy of isolate JKTJ-3 WGC by treatment of seeding substrate.

Inoculation Interval	Treatments	Seed Emergence Rate (%)	Damping-Off Incidence (%)	Seedling Rate (%)	Control Efficacy (%)
0 d	JKTJ-3 WGC	88.3 b ^1^	21.7 c ^1^	69.2 a ^1^	75.6 a ^1^
	Hymexazol WP	91.7 a	40.0 b	52.0 b	55.3 b
	Control	86.7 b	89.3 a	9.8 c	-
1 d	JKTJ-3 WGC	87.5 a	10.0 c	78.8 a	88.6 a
	Hymexazol WP	89.2 a	47.5 b	43.3 b	46.2 b
	Control	82.5 b	88.5 a	10.3 c	-
2 d	JKTJ-3 WGC	90.8 b	0.8 c	90.1 a	99.2 a
	Hymexazol WP	93.3 a	54.2 b	41.6 b	38.3 b
	Control	90.8 b	87.8 a	11.4 c	-
3 d	JKTJ-3 WGC	96.7 a	0.0 c	94.3 a	100.0 a
	Hymexazol WP	97.5 a	59.8 b	38.8 b	30.5 b
	Control	94.3 b	86.3 a	13.3 c	

^1^ The values followed with the same letters within each column for each inoculation interval treatment were not significantly different (*p* > 0.05).

**Table 5 microorganisms-11-01360-t005:** Biocontrol efficacy of isolate JKTJ-3 FC and CF by treatment of seed-soaking.

Seed-Soaking Time	Treatment	Seed Emergence Rate (%)	Damping-Off Incidence (%)	Seedling Rate (%)	Control Efficacy (%)
4 h	JKTJ-3 FC	92.0 bc ^1^	55.7 de ^1^	44.3 c ^1^	39.2 bc ^1^
	JKTJ-3 CF	91.0 c	45.1 f	54.9 a	50.7 a
	Control	95.0 b	91.6 a	8.4 g	
8 h	JKTJ-3 FC	94.0 bc	76.6 b	23.4 ef	14.7 e
	JKTJ-3 CF	95.0 b	50.5 e	49.5 b	43.7 b
	Control	93.0 bc	89.8 a	10.2 g	
12 h	JKTJ-3 FC	95.0 b	72.6 b	27.4 e	22.3 d
	JKTJ-3 CF	99.0 a	63.5 c	36.5 d	32.1 c
	Control	92.0 bc	93.5 a	6.5 g	

^1^ The values followed with the same letters within each column for each seed-soaking trial were not significantly different (*p* > 0.05) according to least significance test.

## Data Availability

The datasets presented in this study can be found in online repositories. The names of the repository/repositories and accession number(s) can be found at: https://www.ncbi.nlm.nih.gov/nuccore/OK271440.

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
