# Peer review of "Efficacy of Streptomyces murinus JKTJ-3 in Suppression of Pythium Damping-Off of Watermelon"

_microorganisms, 2023, doi:10.3390/microorganisms11061360_

Round 1

Reviewer 1 Report

The article deals with some investigations about the biologically-based control of a worldwide diffuse plant pathology using a potentially effective biocontrol microorganism, that is an evergreen interesting subject worth for publication in the journal.

The article is written rather well also if several parts need to be improved and some issues must be properly addressed.

Minor but relevant revisions of the text and the English language are needed.

Some of these revisions are listed in the attached file.

Author Response

Dear reviewer.

We are very greateful to your comments for the manuscript. Based on your comments and requests, we have made extensive modification on the original manuscript. Here, we attached revised manuscript in the formats of both PDF and MS word, for your approval. A document answering every question from the referees was also summarized and enclosed. Please see the attachment. A revised manuscript with the correction sections blue marked was attached as the supplemental material and for easy check/editing purpose. 

Reviewer 2 Report

You need to pay attention to the space between the words and brackets (exp:Line 57,117, etc.).

Why did not You compare the biocontrol efficacy of JKTJ-3 isolate with more than 4 actinomycete isolates?

Author Response

Dear reviewer.

We are very greateful to your comments for the manuscript. Based on your comments and requests, we have made extensive modification on the original manuscript. Here, we attached revised manuscript in the formats of both PDF and MS word, for your approval. A document answering every question from the referees was also summarized and enclosed. Please see the attachment. A revised manuscript with the correction sections highlighted in yellow was attached as the supplemental material and for easy check/editing purpose. If you have any questions, please contact us immediately.

Reviewer 3 Report

The study by Ge et al. provides a comprehensive approach to characterize the biocontrol efficiency of Streptomyces murinus in suppressing watermelon damping-off. The experiments are well designed and presented in detail. Some typing errors were detected, check the spaces between words. More recent references should be added to the manuscript.

Some suggestions for improving the manuscript are listed below:

Lines 80-86: Have these microorganisms been isolated in this study? If so, these lines need to be moved to the Results section. Otherwise, it should be emphasized that they are part of an existing laboratory collection.

Line 87: What is the full species name for C. capsic?

Line 107: How was the percentage of growth inhibition calculated?

Line 202: delete "Streptomyces"

Line 291: This is confusing because actinomycetes are also bacteria; reword the sentence to make it clearer

Line 294: results should be expressed as percent of growth inhibition

Line 307: add p-value at the end of the table description

Line 346: change "website" to "database"

Line 356: since the strain was identified as S. murinus, it should be referred to as S. murinus JKTJ-3; add to the end of the manuscript

Line 474: Add that according to bac dive (https://bacdive.dsmz.de), S. murinus is at risk level 1 of the biosafety level as strongly recommended for biological control agents

Line 507-513: discuss results according to available literature

Line 533: change "antibiotics" to "antifungals"

Lines 545-550: use for conclusion

Line 551: the conclusion should be broadened, summarizing the main findings of the study and include future perspectives

Author Response

Dear reviewer.

We are very greateful to your comments for the manuscript. Based on your comments and requests, we have made extensive modification on the original manuscript. Here, we attached revised manuscript in the formats of both PDF and MS word, for your approval. A document answering every question from the referees was also summarized and enclosed. Please see the attachment. A revised manuscript with the correction sections highlighted in blue was attached as the supplemental material and for easy check/editing purpose. If you have any questions, please contact us immediately.
